# *N*-Hydroxy-*N*-Propargylamide Derivatives of Ferulic Acid: Inhibitors of Cholinesterases and Monoamine Oxidases

**DOI:** 10.3390/molecules27217437

**Published:** 2022-11-01

**Authors:** Óscar M. Bautista-Aguilera, José M. Alonso, Marco Catto, Isabel Iriepa, Damijan Knez, Stanislav Gobec, José Marco-Contelles

**Affiliations:** 1Universidad de Alcalá, Departamento de Química Orgánica y Química Inorgánica, Ctra, Madrid-Barcelona Km 33.6, 28871 Alcalá de Henares, Madrid, Spain; 2Department of Pharmacy-Pharmaceutical Sciences, University of Bari Aldo Moro, Via E. Orabona 4, 70125 Bari, Italy; 3Institute of Chemical Research Andrés M. del Río, Alcalá University, 28805 Alcalá de Henares, Madrid, Spain; 4University of Ljubljana, Faculty of Pharmacy, Askerceva 7, 1000 Ljubljana, Slovenia; 5Laboratory of Medicinal Chemistry (IQOG, CSIC), C/Juan de la Cierva 3, 28006 Madrid, Spain; 6Center for Biomedical Network Research on Rare Diseases (CIBERER), CIBER, ISCIII, 28029 Madrid, Spain

**Keywords:** cholinesterases, ferulic acid, molecular modelling, monoamine oxidase, radical-scavenger, antioxidant

## Abstract

Alzheimer’s disease (AD) is a complex disorder characterized by impaired neurotransmission in cholinergic and monoaminergic neurons, which, in combination with the accumulation of misfolded proteins and increased oxidative stress, leads to the typical features of the disease at the biomolecular level. Given the limited therapeutic success of approved drugs, it is imperative to explore rationally supported therapeutic approaches to combat this disease. The search for novel scaffolds that bind to different receptors and inhibit AD disease-related enzymes could lead to new therapeutic solutions. Here, we describe *N*-hydroxy-*N*-propargylamide hybrids **1**–**6**, which were designed by combining the structures of Contilisant—a multifunctional anti-AD ligand—and ferulic acid, a natural antioxidant with various other biological activities. Among the synthesized compounds, we identified compound **4** as a micromolar inhibitor of hAChE with a potent radical-scavenging capacity comparable to resveratrol and Trolox. In addition, compound **4** chelated copper(II) ions associated with amyloid *β* pathology, mitochondrial dysfunction, and oxidative stress. The promising in vitro activity combined with favorable drug-like properties and predicted blood-brain barrier permeability make compound **4** a multifunctional ligand that merits further studies at the biochemical and cellular levels.

## 1. Introduction

Alzheimer’s disease (AD) is a neurodegenerative disorder, the most common form of senile dementia, and represents one of the most serious public health problems, mainly due to the increasing aging of the population in developed countries [1,2]. AD is characterized by the progressive loss of cognitive abilities, which gradually leads to physical and mental impairment in patients who eventually die from secondary complications related to the disease, such as pneumopathies, septicemia, infected bedsores, or strokes [3]. AD is a multifactorial pathology characterized by two main features: the accumulation of abnormal extracellular deposits of amyloid *β* (A*β*) peptide and intracellular neurofibrillary tangles (NFTs), as well as dramatic neuronal death and a decrease in choline acetyltransferase activity [4]. Oxidative stress and imbalances in the homeostasis of biometals such as Cu, Fe, or Zn [5,6,7] are other important features of AD pathogenesis. Current pharmacotherapy of AD acts mainly on the cholinergic and glutamatergic systems, with the acetylcholinesterase (AChE) and/or butyrylcholinesterase (BChE) inhibitors donepezil [8], galantamine [9], rivastigmine [8], and memantine, an *N*-methyl-d-aspartate antagonist [8,10], being the only drugs clinically used, albeit with limited therapeutic success and benefit. Monoamine oxidases (MAOs) catalyze the oxidation of amines that act as neurotransmitters, releasing H_2_O_2_ and consequently reactive oxygen and nitrogen species [11]. Inhibition of MAO has potent neuroprotective effects by reducing oxidative stress and restoring impaired synaptic plasticity, memory, and learning in a mouse model of AD via control of tonic GABA levels [12]. Parkinson’s disease (PD) is another chronic neurodegenerative disorder affecting 1% of elderly over 60 years of age. PD is characterized by the progressive death of dopaminergic neurons in the substantia nigra and the formation of Lewy bodies containing aggregates of α-synuclein [13]. The deficit in dopaminergic neurotransmission is the basis for the well-known symptoms associated with PD pathology, namely, tremor, rigidity, bradykinesia, and postural instability. Approved therapies for PD increase dopamine levels in the striatum via selective MAO-B inhibition—for example, rasagiline is prescribed as monotherapy in the early stages of PD and as add-on therapy to levodopa in advanced stages of PD [14]. A recently developed reversible MAO-B inhibitor also rescues memory and learning deficits in the APP/PS1 mouse model of AD, which also provides a strong argument for investigating MAO-B inhibitors as AD therapeutics [15]. 

In view of the complex disease pathogenesis, a new therapeutic strategy based on the multitarget small molecule (MSM) approach has been established to design new drugs that can simultaneously interact with different enzymes or receptors involved in AD pathology [16]. In this context, we recently described Contilisant (Figure 1), a blood-brain-barrier (BBB) permeable and highly neuroprotective agent against a number of AD-relevant toxic insults that inhibits neurotransmitter-degrading enzymes (ChE, MAO) or G protein-coupled receptors (H3R, S1R) [17]. Moreover, compared to donepezil, Contilisant has superior in vivo protective effects in Y-maze and radial-arm maze against cognitive decline induced by the *β*-amyloid peptide A*β*_1–42_ oligomers [17]. On the other hand, unlike petroleum-based molecules, natural products such as ferulic acid (FA) (Figure 1) are biologically derived and sustainable substances that are preferred in the chemical and pharmaceutical industries [18,19]. Moreover, they have shown promising pharmacological effects that could be useful for the treatment of AD, such as antioxidant effects, neuroprotection, and promotion of neurogenesis [20]. However, FA has low bioavailability and poor BBB penetration, which limits its further clinical application [21].

Based on these precedents and continuing our contributions to the field, we describe in this preliminary communication the synthesis and biological activity of new hybrids of type **I** molecules (Figure 1) by combining selected functional moieties of FA and Contilisant. The novel *N*-hydroxy[*N*-(prop-2-yn-1-yl)]amide derivatives of FA **1**–**6** (Figure 1) were designed by linking the piperidine motif to the free phenol group of FA via a linear methylene chain, while the carboxylic acid was converted to a *N*-propargylamide, substituted or unsubstituted by a hydroxy motif to impart biometal chelating properties. Among the synthesized compounds, we identified compound **4** with micromolar inhibitory potency against human (h)AChE, antioxidant, and copper(II) chelating properties, making FA analogue **4** a promising lead compound for further development.

## 2. Results

### 2.1. Synthesis

The synthesis of the new FA derivatives **1**–**6** was carried out in short synthetic procedures, as shown in Figure 1, Figure 2, Figure 3 and Figure 4, in good overall yields, starting from readily available precursors. As shown in Figure 1 and as previously described [22], the reaction of FA with MeOH catalyzed with concentrated sulfuric(VI) acid gave ester **7** in quantitative yield, which after standard *O*-alkylation with commercial 1-(2-chloroethyl)piperidine hydrochloride gave ether **8** in 86% yield. Reaction of ether **8** with aqueous hydroxylamine in basic medium gave the target ligand **1** in a modest, unoptimized yield of 30%. A similar protocol, but using 1-(3-chloropropyl)piperidine hydrochloride, gave intermediate **9** in 84% yield and target compound **2** in 34% yield (Figure 1). A slightly different protocol was chosen for the synthesis of compound **3** (Figure 2), which gave better results. Standard *O*-alkylation of compound **7** with commercial bromo-4-chlorobutane and subsequent reaction with piperidine gave the chloride **10** in 80% yield and the intermediate **11** in 86% yield, the reaction of which with hydroxylamine gave the target ligand **3** in 35% yield.

The synthesis of the analogous *N*-propargyl hydroxamate derivatives **4**–**6** was carried out starting from intermediates **8**, **9**, and **11**, respectively, as shown in Figure 3 and Figure 4. As shown in Figure 3, hydrolysis of esters **8** and **9** gave acids **12** (98%) and **13** (96%), respectively. Subsequently, we prepared *N*-(prop-2-yn-1-yl)hydroxylamine hydrochloride as described in [23] and used it in the amidation reaction of **12** and **13** in the presence of HATU and DIPEA to give *N*-propargyl hydroxamates **4** and **5**, respectively. For the preparation of compound **6**, a similar protocol was carried out starting from intermediate **13** via acid **14** (Figure 4). 

It is interesting to note that compounds **4**–**6** are essentially *N*-propargyl hydroxamates (or *N*-hydroxy-*N*-propargyl carboxamide derivatives), which to our knowledge have never been reported in the literature. We were interested in these types of products to evaluate them as potential MAO inhibitors (MAOi) because they possess the *N*-propargyl moiety that can irreversibly bind to the flavin-adenine dinucleotide cofactor of the MAO enzyme [16]. 

In addition, we prepared and biologically assayed simple, non-FA analogues *N*-hydroxybenzamide (**15**) and *N*-hydroxy-*N*-(prop-2-yn-1-yl)benzamide (**16**) (Figure 1). Compound **15** is commercially available and was used as such. Compound **16**, on the other hand, not previously described, was prepared by a reaction of benzoic acid with *N*-(prop-2-yn-1-yl)hydroxylamine hydrochloride (Materials and Methods). All compounds described here had analytical and spectroscopic data that were in good agreement with expected values (see Section 3 and Appendix A) and were therefore subjected to biological evaluation. 

### 2.2. Biological Evaluation: Inhibition of hChEs/hMAOs

Novel compounds **4**–**6**, **15**, and **16** were assayed for their inhibition of hChEs and hMAOs (all from Sigma Aldrich). Inhibition assays were performed according to previously published protocols [24]. All compounds were screened at a concentration of 10 μM, and IC_50_ values were calculated for those showing inhibition *>*60%, by testing seven concentrations ranging from 30–0.01 μM. As shown in Table 1, the compounds were inactive toward hBChE, hMAO-A, and hMAO-B. In contrast, some of them showed modest but selective inhibition of hAChE in the single-digit micromolar range. The most potent inhibitor was **4** (IC_50_ = 2.63 ± 0.57 μM), an *N*-propargylhydroxamate with a piperidine ring linked to oxygen at C4 via an ethylene linker, followed by ligands **3** (IC_50_ = 6.58 ± 0.27 μM) and **6** (IC_50_ = 7.20 ± 0.54 μM), which are nearly equipotent. Compound **3** is a hydroxamate in which the piperidine ring is attached to the C4 oxygen via an *n*-butyl linker, while compound **6** is a corresponding *N*-propargylhydroxamate. Certain conclusions can be drawn regarding the structure-activity relationships (SAR): (i) among hydroxamates **1**–**3**, where the length of the linker connecting the piperidine to the phenyl ring varies, the most potent derivative was analogue **3** with an *n*-butyl linker; (ii) of *N*-propargylhydroxamates **4**–**6**, ligand **4** with a shorter, ethyl spacer was the most potent inhibitor of hAChE; (iii) *N*-propargylhydroxamate **4**, the most potent AChE inhibitor, was surprisingly devoid of any MAO inhibitory activity. 

### 2.3. Molecular Modeling

To understand and predict the binding modes of ferulic acid analogs to hAChE, we performed molecular docking studies with the most potent inhibitor, *N*-propargylhydroxamate **4.**

The structure of hAChE crystallized with fasciculin II (PDB ID: 1B41) was used and docking of compound **4** with the target enzyme was performed using AutoDock Vina software [25], and the results were analyzed using Discovery Studio. From the docking simulation, compound **4** fits into the active site of hAChE and interacts with the amino acid residues of the active site—the catalytic triad and the oxyanionic site at the base of the gorge (Ser203, His447, Gly121 and Gly122) and the peripheral aromatic site (PAS) at the entrance of the gorge (Trp286 and Asp74) (Figure 2).

The ligand-protein interactions of the top binding pose of ligand **4** are shown in Figure 3. The piperidinium moiety of compound **4** interacts in the PAS, where one of the methylene moieties forms an π-alkyl interaction with the aromatic moiety of Tyr72. In addition, the positively charged nitrogen of the piperidine ring forms attractive charge interactions with Asp74 and π-cation interactions with Trp286. In addition, Asp74 is capable of forming carbon-hydrogen bonds with the piperidine and alkyloxy chains. The methoxy group on the benzene ring forms a hydrogen bond with Phe295 (acyl binding pocket), and the benzene ring interacts with Tyr341 through π-π stacking interactions. In addition, the OH of the hydroxamate group forms hydrogen bonds with Gly121, Gly122 (oxyanion hole), and Ser203 (catalytic triad). Finally, the terminal carbon atom of the propargyl group forms π-alkyl interactions with Trp86 (choline-binding pocket), Tyr337, and His447 (part of the AChE catalytic triad). In summary, and based on the docking results, we predict that compound **4** may be a dual-binding site AChE inhibitor that simultaneously interacts with the catalytic active site and PAS of hAChE.

### 2.4. ADME Virtual Profile for Compound **4**

Next, using the QikProp module of the Schrödinger software (QikProp, Schrödinger v.2020-1), we calculated several key ADME properties of the most potent inhibitor identified here, compound **4**.

Because the therapeutic target is in the brain, we first determined whether the potential drug candidate could cross BBB. A QPlogBB predicted value of −0.895 means that the tested compound can cross the BBB (Table 2). Next, we wanted to estimate whether compound **4** is orally bioavailable. According to Lipinski’s rule of five [26], an orally active drug must have no more than one violation of the following criteria: (1) molecular weight less than 500 Daltons (Da), (2) no more than 5 hydrogen bond donors, (3) no more than 10 hydrogen bond acceptors, and (4) an octanol-water partition coefficient log P of less than 5. According to our calculations, this compound could be considered as a potential oral drug because it does not violate any of the above four rules: The molecular weight is 358.436 Da, it has 1.5 hydrogen bond donors, 7.7 hydrogen bond acceptors, and the calculated QPlogPo/w is 2.888. Moreover, the percent oral absorption is 87.124%. To evaluate the bioavailability of the compound, the rule of three of Jorgensen (ROT) was used by estimating the solubility, permeability, and first-pass metabolism in liver according to the following rules: predicted solubility in water (QPlogS) higher than –5.7 mol/dm^3^, predicted apparent permeability in Caco-2 cells (QPPCaco) higher than 22 nm/s, and number of primary metabolites up to 7 [27,28]. According to the predictions, the compound has no violation of ROT. The predicted value for the partition coefficient (QPlogPo/w = 2.888) is within the recommended range (−2–6.5), and the total solvent accessible surface area (SASA = 746.494 Å^2^) is also in a satisfactory range (300.0–1000.0 Å^2^). Water solubility is one of the most important factors affecting drug bioavailability. For a drug to be absorbed, it must first be water soluble and then have the ability to cross biological membranes. The predicted value for water solubility (QPlog S) was also found to be acceptable. Polar surface area plays an important role in drug absorption, i.e., a polar surface area (PSA) greater than 200 Å^2^, should be avoided. Compound **4** has a PSA of 67.634 Å^2^, suggesting good absorption. In vitro Caco-2 cell permeability is an indicator of intestinal absorption of drugs. According to our results, this ligand shows moderate permeation of 261.504 nm/s. Other physicochemical descriptors determined via QikProp (Table 2) are in the acceptable range for human use, confirming that compound **4** is a drug-like molecule and a possible CNS drug.

### 2.5. Antioxidative and Metal Chelating Properties

Increasing evidence on the pathology of AD suggests that oxidative stress plays a crucial role through several pathways [29]. Several antioxidants such as resveratrol and curcumin are being extensively studied in the context of AD therapy, and studies suggest that both may delay cellular senescence and aging [30]. In addition, antioxidants or natural products containing polyphenols are also being investigated in clinical trials involving AD patients [31]. Therefore, the antioxidant properties of compound **4** were investigated using the 2,2-diphenyl-1-picrylhydrazyl radical (DPPH) scavenging assay, a simple and reliable in vitro assay [32]. *N*-Propargyhydroxamate **4** was a modest radical scavenger, with an EC_50_ = 40.9 ± 0.5 µM, a value lower than that of resveratrol (EC_50_ = 50.1 ± 4.6 µM), but 3.3-fold higher than that of Trolox (EC_50_ = 12.4 ± 0.3 µM). Hydroxamic acids are considered to be potent antioxidants [33], and, in combination with cholinesterase inhibitory activity, could not only alleviate symptoms but also have disease-modifying effects [34].

Another therapeutic possibility currently under investigation is metal chelation, which could affect not only the redox cycling of metal ions but also A*β*- and tau-pathology [35]. Metal chelation of compound **4** was investigated using a UV-Vis absorption assay, in which the compound was incubated with equimolar concentrations of the major bivalent biometals (i.e., Ca^2+^, Mg^2+^, Zn^2+^, Cu^2+^, Fe^2+^, Mn^2+^, Co^2+^, and Ni^2+^) (Figure 4). As indicated by the spectral change and the presence of an isosbestic point in the presence of Cu^2+^ ions, compound **4** efficiently bound Cu^2+^ ions, which are at the center of vicious oxidative stress, A*β* aggregation, and several other AD-related pathological changes [36]. A slight change in the presence of Fe^2+^ ions also indicated that ferrous ions were also bound by compound **4**, which is not surprising since Desferrioxamine B, a hydroxamate-based siderophore of bacterial origin, efficiently binds ferrous and ferric ions [37]. A subtle spectral change was also observed in the presence of Ca^2+^ ions (Figure 4A), but the absence of an isosbestic point indicates that this spectral change does not appear to be due to the formation of a Ca^2+^/**4** complex.

## 3. Materials and Methods

All reagents were purchased from Sigma Aldrich. Melting points were determined using a Kofler apparatus (Wagner Munz, Munich, Germany). The progress of the reactions was monitored by TLC on aluminum plates with silica gel 60 F254 from Merck (Kenilworth, NJ, USA). ^1^H and ^13^C-NMR spectra were recorded on a Bruker spectrometer operating at 500, 400, 300, and 100 MHz; DMSO-*d*_6_, chloroform-d, D_2_O, and CD_3_OD were used as solvents. Chemical shifts are given in parts per million (ppm), using tetramethylsilane (TMS) as an internal reference. The multiplicities of the signals are given with the following abbreviations: s, singlet; d, doublet; t, triplet; q, quadruplet; and m, multiplet; the coupling constants are given in Hz. Elemental analyses were performed using a Carlo Erba EA 1108 analyzer, and analytical results were within ± 0.2% of theoretical values for all compounds.

### 3.1. Synthesis of Compounds **1**–**6**

General method for *O*-Alkylation (A). To a solution of methyl (*E*)-3-(4-hydroxy-3-methoxyphenyl)acrylate (1 equiv) in CHCl_3_/H_2_O (30 mL, 12.5 M, vol/vol, 5/1), and the corresponding commercial alkyl halide (1.5 equiv), K_2_CO_3_ (3 equiv) was added. Then, the reaction mixture was stirred overnight at 80 °C. The mixture was cooled, extracted with DCM/H_2_O, and then purified using flash column chromatography to afford the corresponding product.

General method for synthesis of hydroxamates (B). To a solution of NaOH (4 equiv) in NH_2_OH 50% (aq.) (28 equiv) cooled to 0 °C, a solution of the corresponding ester (1 equiv) in THF (0.4 M) was added dropwise. The reaction mixture was warmed up to room temperature (rt) and stirred for 1 h at rt. The solvent was evaporated under reduced pressure and the pH of the mixture was lowered to 7–8 with HCl (2 M), extracted with EtOAc (3 × 100 mL), dried, filtered, and evaporated. The crude product was purified via flash column chromatography. 

General method for the hydrolysis of esters (C). To a solution of the ester (1 equiv) in MeOH (0.3 M), a solution of KOH (5 equiv) in distilled water (6.0 M) was added dropwise. The reaction mixture was stirred for 4 h at rt. After completion of the reaction, the pH of the mixture was adjusted to about 2, and then the solvent was evaporated under reduced pressure. The crude product was suspended in EtOH (1.0 M) and stirred under reflux for 15 min, filtered, the solvent was evaporated, and the crude product was recrystallized to give the pure acid.

General method for the synthesis of the *N*-propargylhydroxamates (D). To a solution of the corresponding acid in dry DMF (0.6 M), HATU (1.0 equiv) and DIPEA (2.5 equiv) were added. The reaction mixture was stirred at rt for 15 min. *N*-(Prop-2-yn-1-yl)hydroxylamine hydrochloride (1.0 equiv) and another equivalent of DIPEA (1.0 equiv) were added. The reaction mixture was stirred overnight at rt. The solvent was evaporated under reduced pressure, and the crude product was purified via column chromatography (DCM/MeOH 0–5%), with MeOH containing 10% of NH_4_OH, followed by recrystallization from Et_2_O/MeOH to obtain the pure product.

*Methyl (E)-3-(4-hydroxy-3-methoxyphenyl)acrylate (**7**)* [22]. To a solution of ferulic acid (10.0 g, 51.5 mmol) in MeOH (30 mL), cc H_2_SO_4_ (1.3 mL) was added. Then, the reaction mixture was stirred at reflux for 2 h. The reaction mixture was cooled, quenched with a saturated solution of NaHCO_3_, and extracted with EtOAc (3 × 250 mL). The organic layer was dried over Na_2_SO_4_, filtered, and concentrated under reduced pressure to obtain compound **7** (10.5 g) [22], in quantitative yield as a pale pink solid: ^1^H NMR (400 MHz, CDCl_3_) δ 7.62 (d, *J =* 15.9 Hz, 1H, Ar*CH*=CH), 7.09–6.97 (m, 2H, ArH), 6.91 (d, *J =* 8.1 Hz, 1H, ArH), 6.29 (d, *J =* 15.9 Hz, 1H, ArCH=*CH*), 5.94 (s, 1H, OH), 3.92 (s, 3H, OCH_3_), 3.79 (s, 3H, CO_2_CH_3_); ^13^C NMR (101 MHz, CDCl_3_) δ 167.77, 148.02, 146.81, 144.99, 126.95, 123.04, 115.15, 114.77, 109.40, 55.94, 51.63.

*Methyl (E)-3-(3-methoxy-4-(2-(piperidin-1-yl)ethoxy)phenyl)acrylate (**8**)*. Following general method A, a solution of compound **7** (2.0 g, 9.62 mmol) in CHCl_3_:H_2_O (30 mL) and 1-(2-chloroethyl)piperidine hydrochloride (2.65 g, 14,43 mmol) was treated with K_2_CO_3_ (3.98 g, 28.86 mmol). After complete reaction, workup, and purification (DCM/MeOH 0–5%, MeOH containing 10% NH_4_OH), compound **8** (2.64 g, 86%) was isolated as a solid: mp 50–53 °C; IR (KBr) υ 2933, 2851, 2784, 1597, 1709, 1634, 1509, 1252, 1138, 1030 cm^−1^; ^1^H NMR (400 MHz, CDCl_3_) δ 7.64 (d, *J* = 16.0, 2.7 Hz, 1H), 7.13–7.02 (m, 2H), 6.89 (d, *J* = 8.3, 2.7 Hz, 1H), 6.32 (d, *J* = 16.0, 2.7 Hz, 1H), 4.19 (t, *J* = 6.4 Hz, 2H), 3.90 (s, 3H), 3.81 (s, 3H), 2.88–2.79 (m, 2H), 2.58–2.49 (m, 4H), 1.69–1.58 (m, 4H), 1.52–1.40 (m, 2H); ^13^C NMR (101 MHz, CDCl_3_) δ 167.7, 150.5, 149.5, 144.8, 127.5, 122.5, 115.5, 112.5, 110.1, 66.5, 57.5 (2C), 55.9, 55.0, 51.6, 25.8 (2C), 24.1. HRMS (ESI): Calcd for C_18_H_25_NO_4_: 319.1784. Found: 319.179. 

*(E)-N-Hydroxy-3-(3-methoxy-4-(2-(piperidin-1-yl)ethoxy)phenyl)acrylamide (**1**).* Following general method B, a solution of NaOH (0,2 g, 4.96 mmol) in NH_2_OH (aq.) (2.2 mL, 34.72 mmol) at 0 °C was reacted with a solution of compound **8** (0.4 g, 1.25 mmol) in THF (4 mL). After complete reaction, workup, purification (DCM-MeOH 0–50%, MeOH containing 10% of NH_4_OH), and recrystallization from CHCl_3_, compound **1** (0.120 g, 30%) was isolated: mp 127–130 °C; IR (KBr) *v* 3546, 3166, 2937, 1668, 1626, 1259, 1142, 1051, 1030 cm^−1^; ^1^H NMR (500 MHz, CD_3_OD) δ 7.50 (d, *J* = 15.7 Hz, 1H, Ar*CH*=CH), 7.15 (d, *J* = 2.0 Hz, 1H, H-2), 7.11 (dd, *J* = 8.3, 2.0 Hz, 1H, H-6), 6.97 (d, *J* = 8.3 Hz, 1H, H-5), 6.34 (d, *J* = 15.7 Hz, 1H, ArCH=*CH*), 4.18 (t, *J* = 5.7 Hz, 2H, CH_2_O), 3.86 (s, 3H, OCH_3_), 2.82 (t, *J* = 5.7 Hz, 2H, *CH_2_*CH_2_O), 2.60 [br s, 4H, N(CH_2_)_2_], 1.64 [m, *J* = 5.6 Hz, 4H, N(CH_2_)_2_(*CH_2_*)_2_)], 1.49 [q, *J* = 5.9 Hz, 2H, N(CH_2_)_2_(CH_2_)_2_*CH_2_*] (signals for RCON*H*O*H* were not detected); ^13^C NMR (126 MHz, CD_3_OD) δ 165.3 (C=O), 149.9 (C-4), 149.8 (C-3), 140.1 (Ar*CH*=CH), 128.3 (C-1), 121.5 (C-6), 114.9 (ArCH=*CH*), 113.1 (C-5), 110.3 (C-2), 66.4 (CH_2_O), 57.3 (*CH_2_*CH_2_O), 55.0 (OCH_3_), 54.5 [2C, N(CH_2_)_2_], 25.0 [2C, N(CH_2_)_2_(*CH_2_*)_2_], 23.5 [N(CH_2_)_2_(CH_2_)_2_*CH_2_*]. HRMS (ESI): Calcd for C_17_H_24_N_2_O_4_: 320.1739. Found: 320.1736. 

*Methyl (E)-3-(3-methoxy-4-(3-(piperidin-1-yl)propoxy)phenyl)acrylate (**9**).* Following general method A, a solution of compound **7** (3.0 g, 14.4 mmol), 1-(3-chloropropyl)piperidine hydrochloride (4.41 g, 21.6 mmol) in CHCl_3_:H_2_O (60 mL) was treated with K_2_CO_3_ (5.97 g, 43.3 mmol). After complete reaction, workup, and purification (DCM-MeOH 0–5%, MeOH containing 10% of NH_4_OH), compound **9** (4.03 g, 84%) was obtained as a solid: mp 55–58 °C; IR (KBr) υ 2935, 2851, 2767, 1716, 1634, 1597, 1511, 1466, 1258, 1162, 1138 cm^−1^; ^1^H NMR (400 MHz, CDCl_3_) δ 7.65 (d, *J* = 15.8 Hz, 1H), 7.17–7.01 (m, 2H), 6.91 (d, *J* = 8.5 Hz, 1H), 6.32 (d, *J* = 15.8 Hz, 1H), 4.13 (t, *J* = 6.9 Hz, 2H), 3.91 (s, 3H), 3.82 (s, 3H), 2.49 (t, *J* = 7.2 Hz, 2H), 2.47–2.31 (m, 4H), 2.05 (t, *J* = 7.1 Hz, 2H), 1.69–1.56 (m, 4H), 1.52–1.37 (m, 2H); ^13^C NMR (101 MHz, CDCl_3_) δ 167.7, 150.7, 149.5, 144.9, 127.3, 122.6, 115.3, 112.6, 110.2, 67.6, 56.0 (2C), 55.7, 54.6, 51.6, 26.6, 26.0 (2C), 24.4. HRMS (ESI): Calcd for C_19_H_27_NO_4_: 333.1940. Found: 333.1945.

*(E)-N-Hydroxy-3-(3-methoxy-4-(3-(piperidin-1-yl)propoxy)phenyl)acrylamide (**2**).* Following general method B, a solution of NaOH (0,432 g, 10.81 mmol) in NH_2_OH (aq.) (2.34 mL, 75.6 mmol) was reacted with a solution of compound **8** (0.9 g, 2.7 mmol) in THF (5 mL). After complete reaction, workup, purification (DCM-MeOH 0–50%, MeOH containing 10% of NH_4_OH), and recrystallization from ether/MeOH, compound **2** (0.325 g, 34%) was obtained as a solid: mp: 160–162 °C; IR (KBr) *v* 3191, 2976, 2949, 2824, 1650, 1602, 1269, 1129, 1021 cm^−1^; ^1^H NMR (500 MHz, CD_3_OD) δ 7.49 (d, *J* = 15.7 Hz, 1H, Ar*CH*=CH), 7.14 (d, *J* = 1.7 Hz, 1H, H-2), 7.11 (dd, *J* = 8.3, 1.7 Hz, 1H, H-6), 6.96 (d, *J* = 8.3 Hz, 1H, H-5), 6.34 (d, *J* = 15.7 Hz, 1H, ArCH=*CH*), 4.07 (t, *J* = 6.2 Hz, 2H, CH_2_O), 3.86 (s, 3H, OCH_3_), 2.62–2.54 (m, 2H, *CH_2_*CH_2_CH_2_O), 2.51 [br s, 4H, N(CH_2_)_2_], 2.08–1.95 (m, 2H, *CH_2_*CH_2_O), 1.68–1.59 [m, 4H, N(CH_2_)_2_(*CH_2_*)_2_)], 1.50 [br s, 2H, N(CH_2_)_2_(CH_2_)_2_*CH_2_*] (signals for RCON*H*O*H* were not detected); ^13^C NMR (126 MHz, CD_3_OD) δ 168.2 (C=O), 150.2 (C-4), 149.7 (C-3), 140.1 (Ar*CH*=CH), 128.1 (C-1), 121.6 (C-6), 114.8 (ArCH=*CH*), 112.8 (C-5), 110.3 (C-2), 67.0 (CH_2_O), 55.6 (*CH_2_*CH_2_CH_2_O), 55.1 (OCH_3_), 54.0 [2C, N(CH_2_)_2_], 25.9 (*CH_2_*CH_2_O), 25.0 [2C, N(CH_2_)_2_(*CH_2_*)_2_], 23.7 [N(CH_2_)_2_(CH_2_)_2_*CH_2_*]; HRMS (ESI): Calcd. for C_18_H_26_N_2_O_4_: 334.1909. Found: 334.1893.

*Methyl (E)-3-(4-(4-chlorobutoxy)-3-methoxyphenyl)acrylate (**10**).* To a suspension of compound **7** (1.0 g, 4.80 mmol) and K_2_CO_3_ (3.97 g, 28.8 mmol) in dry CH_3_CN (20 mL), 1-bromo-4-chlorobutane (1.67 mL, 14.4 mmol) was added dropwise. The reaction mixture was stirred to reflux for 2 h, cooled, filtered, and extracted with DCM/H_2_O. The organic layer was evaporated under reduced pressure and purified via flash column chromatography (DCM/MeOH 0–10%) to get compound **10** (1.14 g, 80%) as an oil: IR (KBr) *v* 2948, 2871, 1701, 1597, 1507, 1433, 1306, 1248, 1135 cm^−1^; ^1^H NMR (400 MHz, CDCl_3_) δ 7.65 (d, *J* = 15.9 Hz, 1H), 7.15–7.01 (m, 2H), 6.87 (d, *J* = 9.1 Hz, 1H), 6.33 (dd, *J* = 15.9, 2.3 Hz, 1H), 4.11 (br s, 2H), 3.91 (s, 3H), 3.82 (s, 3H), 3.66 (d, *J* = 5.8 Hz, 2H), 2.07–1.95 (m, 4H); ^13^C NMR (101 MHz, CDCl_3_) δ 167.7, 150.5, 149.6, 144.7, 127.5, 122.5, 115.5, 112.5, 110.2, 68.10, 56.0, 51.6, 44.7, 29.2, 26.5. HRMS (ESI): Calcd for C_15_H_19_ClO_4_: 298.0972. Found: 298.097.

*Methyl (E)-3-(3-methoxy-4-(4-(piperidin-1-yl)butoxy)phenyl)acrylate (**11**).* To a solution of compound **10** (1.14 g, 3.825 mol) in dry CH_3_CN (20 mL), KI (1.3 g, 7.65 mmol) was added, and the reaction mixture was stirred at reflux for 4 h. Then, the mixture was cooled at rt, and K_2_CO_3_ (2.11 g, 15.3 mmol) and piperidine (1.13 mL, 11.5 mmol) were added. The reaction mixture was stirred at reflux overnight. Then, the mixture was cooled, the excess of K_2_CO_3_ filtered, and the solution was extracted with DCM/H_2_O. The organic layer was dried, evaporated under reduced pressure, and purified via flash column chromatography (DCM/MeOH 0–2%, MeOH containing 10% of NH_4_OH) to get compound **11** (1.16 g, 86%) as an oil: IR (KBr) *v* 2933, 2722, 2532, 1707, 1634, 1597, 1511, 1254, 1162 cm^−1^; ^1^H NMR (400 MHz, CDCl_3_) δ 7.63 (d, *J* = 16.7 Hz, 1H), 7.09 (d, *J* = 8.4 Hz, 1H), 7.05 (br s, 1H), 6.87 (d, *J* = 8.8 Hz, 1H), 6.32 (d, *J* = 16.7 Hz, 1H), 4.09 (t, *J* = 6.0 Hz, 2H), 3.90 (s, 3H), 3.81 (s, 3H), 2.99–2.80 (m, 6H), 2.07–1.99 (m, 2H), 1.98–1.90 (m, 6H), 1.64 (br s, 2H); ^13^C NMR (101 MHz, CDCl_3_) δ 167.6, 150.3, 149.4, 144.7, 127.6, 122.5, 115.6, 112.7, 110.1, 68.4, 57.8 (2C), 55.9, 53.6, 51.6, 26.7 (2C), 23.6, 22.9, 21.9. HRMS (ESI): Calcd for C_20_H_29_NO_4_: 347.2097. Found: 347.2097.

*(E)-N-Hydroxy-3-(3-methoxy-4-(4-(piperidin-1-yl)butoxy)phenyl)acrylamide (**3**).* Following general method B, a solution of NaOH (0.23 g, 5.76 mmol) in NH_2_OH (aq.) (2.5 mL, 40.3 mmol) cooled at 0 °C was reacted with a solution of **11** (0.5 g, 1.44 mmol) in THF (4 mL). After complete reaction, workup, purification (DCM-MeOH 0–50%, MeOH containing 10% of NH_4_OH), and recrystallization from CHCl_3_, compound **3** (0.175 g, 35%) was isolated as a white solid: mp 107–110 °C; IR (KBr) *v* 3186, 2935, 2850, 1655, 1618, 1513, 1422, 1264, 1140, 1050, 1033, 1007 cm^−1^; ^1^H NMR (500 MHz, CD_3_OD) δ 7.49 (d, *J* = 15.7 Hz, 1H, Ar*CH*=CH), 7.14 (d, *J* = 1.8 Hz, 1H, H-2), 7.10 (dd, *J* = 8.3, 1.8 Hz, 1H, H-6), 6.95 (d, *J* = 8.3 Hz, 1H, H-5), 6.33 (d, *J* = 15.7 Hz, 1H, ArCH=*CH*), 4.06 (t, *J* = 6.1 Hz, 2H, CH_2_O), 3.86 (s, 3H, OCH_3_), 2.50 [br s, 4H, N(CH_2_)_2_], 2.48-2.41 (m, 2H, *CH_2_*CH_2_CH_2_O), 1.84-1.77 (m, 2H, *CH_2_*CH_2_O), 1.77-1.71 (m, 2H, *CH_2_*CH_2_CH_2_CH_2_O), 1.66-1.60 [m, 4H, N(CH_2_)_2_(*CH_2_*)_2_], 1.53-1.45 [m, 2H, N(CH_2_)_2_(CH_2_)_2_*CH_2_*] (signals for RCON*H*O*H* were not detected); ^13^C NMR (126 MHz, CD_3_OD) δ 165.4 (C=O), 150.3 (C-4), 149.6 (C-3), 140.0 (Ar*CH*=CH), 127.9 (C-1), 121.6 (C-6), 114.8 (ArCH=*CH*), 112.7 (C-5), 110.2 (C-2), 68.4 (CH_2_O), 58.6 (*CH_2_*CH_2_CH_2_O), 55.01 (OCH_3_), 54.0 [2C, N(CH_2_)_2_], 26.9 (*CH_2_*CH_2_O), 24.9 [2C, N(CH_2_)_2_(*CH_2_*)_2_], 23.7 [N(CH_2_)_2_(CH_2_)_2_*CH_2_*], 22.6 (*CH_2_*CH_2_CH_2_CH_2_O). HRMS (ESI): Calcd for C_19_H_28_N_2_O_4_: 348.2046. Found: 348.2049.

*(E)-3-(3-Methoxy-4-(2-(piperidin-1-yl)ethoxy)phenyl)acrylic acid (**12**).* Following general method C, a solution of compound **8** (2.0 g, 6.26 mmol) in MeOH (15 mL) was treated with a solution of KOH (1.75 g, 31.34 mmol) in distilled water (5 mL). After complete reaction, workup, and recrystallization, compound **12** (1.87 g, 98%) was a solid: mp 140–143 °C; IR (KBr) *v* 3404, 2942, 2648, 2536, 1697, 1597, 1511, 1455, 1258, 1138 cm^−1^; ^1^H NMR (300 MHz, CD_3_OD) δ 7.35 (d, *J* = 15.9 Hz, 1H), 7.17 (d, *J* = 1.9 Hz, 1H), 7.06 (dd, *J* = 8.3, 1.9 Hz, 1H), 6.98 (d, *J* = 8.3 Hz, 1H), 6.37 (d, *J* = 15.9 Hz, 1H), 4.33 (t, *J* = 5.1 Hz, 2H), 3.89 (s, 3H), 3.41 (t, *J* = 5.1 Hz, 2H), 3.29–3.17 (m, 4H), 1.90–1.78 (m, 4H), 1.71–1.60 (m, 2H) (signal for RCO_2_*H* was not detected); ^13^C NMR (101 MHz, CD_3_OD) δ 169.1, 149.8, 148.9, 144.5, 129.4, 122.1, 116.7, 114.3, 110.5, 63.3, 55.8 (2C), 55.2, 53.6, 22.8 (2C), 21.2. HRMS (ESI): Calcd for C_17_H_23_NO_4_: 305.1627. Found: 305.1637.

*(E)-3-(3-Methoxy-4-(3-(piperidin-1-yl)propoxy)phenyl)acrylic acid (**13**).* Following general method C, a solution of compound **9** (2.0 g, 6.0 mmol) in MeOH was reacted with a solution of KOH (1.68 g, 30.0 mmol) in distilled water (5 mL). After complete reaction, workup, and recrystallization, compound **13** (1.83 g, 96%) was isolated as a white solid: mp 85–88 °C; IR (KBr) *v* 3386, 2942, 2640, 2542, 1697, 1630, 1597, 1453, 1258, 1138 cm^−1^; ^1^H NMR (300 MHz, DMSO-d6) δ 12.21 (br s, 1H), 10.53 (br s, 1H), 7.51 (d, *J* = 15.9 Hz, 1H), 7.32 (d, *J* = 1.9 Hz, 1H), 7.18 (dd, *J* = 8.4, 1.9 Hz, 1H), 6.99 (d, *J* = 8.4 Hz, 1H), 6.44 (d, *J* = 15.9 Hz, 1H), 4.07 (t, *J* = 6.1 Hz, 2H), 3.80 (s, 3H), 3.50–3.37 (m, 2H), 3.12 (br t, *J* = 8.0 Hz, 2H), 2.85 (br s, 2H), 2.30–2.13 (m, 2H), 1.87–1.65 (m, 5H), 1.37 (br s, 1H). HRMS (ESI): Calcd for C_18_H_25_NO_4_: 319.1784. Found: 319.1786. 

*N-(Prop-2-yn-1-yl)hydroxylamine hydrochloride* [23]*. N*,*O*-Di-boc hydroxylamine (2.0 g, 8.58 mmol) was dissolved in dry DMF (0.12 M) under an argon atmosphere and K_2_CO_3_ (1.54 g, 11.15 mmol) was added. Propargyl bromide (80% solution in toluene, 1.05 mL, 9.44 mmol) was added dropwise, and the reaction mixture was stirred overnight. Once the reaction was done, the solvent was evaporated under reduced pressure and purified via flash chromatography to obtain a colorless oil (2.17 g) that was dissolved in EtOAc (0.2 M) and distilled water (35 mL) and treated with concentrated HCl (35 mL) via dropwise addition. The reaction mixture was stirred for 5 h. The solvents were evaporated in vacuo to give *N*-(prop-2-yn-1-yl)hydroxylamine hydrochloride as a brown solid (0.82 g, 95%). ^1^H NMR (400 MHz, D_2_O) δ 4.14 (d, *J* = 2.6 Hz, 2H), 2.98 (t, *J* = 2.6 Hz, 1H); ^13^C NMR (75 MHz, MeOD) δ 79.4, 72.8, 42.1. HRMS (ESI^+^) m/z Calcd for C_3_H_6_NO (M+H)^+^: 72.0449. Found: 72.0436 [23]. 

*(E)-N-Hydroxy-3-(3-methoxy-4-(2-(piperidin-1-yl)ethoxy)phenyl)-N-(prop-2-yn-1-yl)acrylamide (**4**).* Following general method D, a solution of compound **12** (1.13 g, 3.31 mmol) in dry DMF (5 mL) was reacted with HATU (1.25 g, 3.31mmol) and DIPEA (2 mL, 11.58 mmol). The reaction mixture was stirred at rt for 15 min. Next, *N*-(prop-2-yn-1-yl)hydroxylamine hydrochloride (0.355 g, 3.31 mmol) and an extra equivalent of DIPEA (0.57 mL, 3.31 mmol) were added. After complete reaction, workup, purification via chromatography and recrystallization, compound **4** (0.332 g, 28%) was obtained as a solid: mp 74–76 °C; IR (KBr) *v* 3437, 3270, 2941, 2832, 1647, 1608, 1510, 1260, 1206, 1031 cm^−1^. ^1^H NMR (500 MHz, CD_3_OD) δ 7.56 (d, *J* = 15.8 Hz, 1H, Ar*CH*=CH), 7.26–7.11 (m, 3H), 6.99 (d, *J* = 15.8 Hz, 1H, ArCH=*CH*), 4.49 (d, *J* = 2.4 Hz, 2H, N(OH)C*H_2_*C≡CH), 4.19 (t, *J* = 5.7 Hz, 2H, CH_2_O), 3.87 (s, 3H, OCH_3_), 2.84 (t, *J* = 5.7 Hz, 2H, *CH_2_*CH_2_O), 2.66 (t, *J* = 2.5 Hz, 1H, N(OH)CH_2_C≡C*H*), 2.61 [br s, 4H, N(CH_2_)_2_], 1.64 [p, *J* = 5.6 Hz, 4H, N(CH_2_)_2_(*CH_2_*)_2_)], 1.50 (q, *J* = 5.8 Hz, 2H, N(CH_2_)_2_(CH_2_)_2_*CH_2_*] (signal for N(O*H*)CH_2_C≡CH was not detected); ^13^C NMR (126 MHz, CD_3_OD) δ 167.4, 150.2 (C-4), 149.8 (C-3), 143.1 (Ar*CH*=CH), 128.5 (C-1), 122.0 (C-6), 113.6 (ArCH=*CH*), 113.1 (C-5), 110.6 (C-2), 77.4 (NOH*C*≡CH), 71.6 (NOHC≡*CH*), 66.4 (CH_2_O), 57.3 (*CH_2_*CH_2_O), 55.1 (OCH_3_), 54.5 [2C, N(CH_2_)_2_], 37.7 (*CH_2_*C≡CH), 25.0 [2C, N(CH_2_)_2_(*CH_2_*)_2_], 23.5 [N(CH_2_)_2_(CH_2_)_2_*CH_2_*]. HRMS (ESI) Calcd. for C_20_H_26_N_2_O_4_: 358.1905. Found 358.1893.

*(E)-N-Hydroxy-3-(3-methoxy-4-(3-(piperidin-1-yl)propoxy)phenyl)-N-(prop-2-yn-1-yl)acrylamide (**5**).* Following general method D, a solution of compound **13** (0.89 g, 2.5 mmol) in dry DMF (4 mL) was reacted with HATU (0.952 g, 2.5 mmol) and DIPEA (1.52 mL, 8.75 mmol). The reaction mixture was stirred at rt for 15 min. Subsequently, *N*-(prop-2-yn-1-yl)hydroxylamine hydrochloride (0.268 g, 2.5 mmol) and an extra equivalent of DIPEA (0.43 mL, 2.5 mmol) were added. After complete reaction, workup, purification via chromatography and recrystallization, compound **5** (0.280 g, 30%) was isolated as a solid: mp 156–158 °C; IR (KBr) *v* 3294, 2934, 2849, 1646, 1594, 1517, 1260, 1137, 1044 cm^−1^; ^1^H NMR (500 MHz, CD_3_OD) δ 7.56 (d, *J* = 15.8 Hz, 1H, Ar*CH*=CH), 7.22–7.11 (m, 3H), 6.97 (d, *J* = 15.8, Hz, 1H, ArCH=*CH*), 4.49 (d, *J* = 2.3 Hz, 2H, N(OH)C*H_2_*C≡CH), 4.08 (t, *J* = 6.2 Hz, 2H, CH_2_O), 3.87 (s, 3H, OCH_3_), 2.65 (t, *J* = 2.5 Hz, 1H, N(OH)CH_2_C≡C*H*), 2.61–2.56 (m, 2H, C*H_2_*CH_2_CH_2_O), 2.52 (br s, 4H, N(CH_2_)_2_), 2.07–1.98 (m, 2H, C*H_2_*CH_2_O), 1.64 [dt, *J* = 11.2, 5.7 Hz, 4H, N(CH_2_)_2_(*CH_2_*)_2_)], 1.50 (d, *J* = 5.2 Hz, 2H, N(CH_2_)_2_(CH_2_)_2_*CH_2_*] (signal for N(O*H*)CH_2_C≡CH was not detected); ^13^C NMR (126 MHz, CD_3_OD) δ 167.5, 150.4 (C-4), 149.6 (C-3), 143.0 (Ar*CH*=CH), 128.2 (C-1), 122.1 (C-6), 113.5 (C-5), 112.7 (ArCH=*CH*), 110.5 (C-2), 77.4 (NOH*C*≡CH), 71.6 (NOHC≡*CH*), 67.0 (CH_2_O), 55.6 (C*H_2_*CH_2_CH_2_O), 55.1 (OCH_3_), 54.0 [2C, N(CH_2_)_2_], 37.8 (*CH_2_*C≡CH), 25.8 (C*H_2_*CH_2_O), 25.0 [2C, N(CH_2_)_2_(*CH_2_*)_2_)], 23.7 [N(CH_2_)_2_(CH_2_)_2_*CH_2_*]. HRMS (ESI) Calcd. for C_21_H_28_N_2_O_4_: 372.2035. Found 372.2049.

*(E)-3-(3-Methoxy-4-(4-(piperidin-1-yl)butoxy)phenyl)acrylic acid (**14**).* Following the general method C, a solution of compound **11** (2.40 g, 6.92 mmol) in MeOH (23 mL) was treated with a solution of KOH (1.94 g, 34.6 mmol) in H_2_O (5 mL). After complete reaction, workup, and recrystallization, compound **14** (1.10 g, 48%) was obtained as a white solid: mp 185–187 °C; IR (KBr) *v* 3418, 2946, 2646, 2543, 1697, 1632, 1597, 1511, 1261, 1138 cm^−1^; ^1^H NMR (300 MHz, DMSO-*d*_6_) δ 12.19 (bs, 1H), 9.89 (bs, 1H), 7.50 (d, *J* = 15.9 Hz, 1H), 7.31 (s, 1H), 7.18 (d, *J* = 8.1 Hz, 1H), 6.97 (d, *J* = 8.3 Hz, 1H), 6.43 (d, *J* = 15.9 Hz, 1H), 4.02 (t, *J* = 6.1 Hz, 2H), 3.81 (s, 3H), 3.48–3.39 (m, 2H), 3.07 (bs, 2H), 2.81 (bs, 2H), 1.90–1.59 (m, 10H). HRMS (ESI): Calcd for C_19_H_27_NO_4_: 333.1940. Found: 333.1945.

*(E)-N-Hydroxy-3-(3-methoxy-4-(4-(piperidin-1-yl)butoxy)phenyl)-N-(prop-2-yn-1-yl)acrylamide (**6**).* Following general method D, a solution of compound **14** in dry DMF (5 mL) was reacted with HATU (1.16 g, 2.98 mol) and DIPEA (1.8 mL, 10.43 mmol). The reaction mixture was stirred at rt for 15 min; then, *N*-(prop-2-yn-1-yl)hydroxylamine hydrochloride (0.319 g, 2.98 mmol) and an extra equivalent of DIPEA (0.77 mL, 4.47 mmol) were added. After complete reaction, workup, purification via chromatography and recrystallization, compound **6** (0.344 g, 27%) was isolated: mp 84–86 °C; IR (KBr) *v* 3423, 3254, 2933, 2835, 2120, 1646, 1595, 1509, 1424, 1214, 1049, 1036 cm^−1^; ^1^H NMR (400 MHz, CD_3_OD) δ 7.57 (d, *J* = 15,8 Hz, 1H, Ar*CH*=CH), 7.18 (m, 3H), 6.98 (d, *J* = 15.8 Hz, 1H, ArCH=*CH*), 4.51 (d, *J* = 2.8 Hz, 2H, N(OH)C*H_2_*C≡CH), 4.08 (t, *J* = 5.8 Hz, 2H, CH_2_O), 3.89 (s, 3H, OCH_3_), 2.67 (t, *J* = 2.5 Hz, 1H, N(OH)CH_2_C≡C*H*), 2,54 [m, 4H, N(CH_2_)_2_], 2.51 (m, 2H, C*H*_2_CH_2_CH_2_CH_2_O), 1.82 (m, 2H, C*H_2_*CH_2_O), 1.79–1.73 (m, 2H, C*H*_2_CH_2_CH_2_O), 1.65 [q, *J* = 5.8 Hz, 4H, N(CH_2_)_2_(*CH_2_*)_2_], 1.52 (br s, 2H, N(CH_2_)_2_(CH_2_)_2_*CH_2_*) (signal for N(O*H*)CH_2_C≡CH was not detected); ^13^C NMR (101 MHz, CD_3_OD) δ 167.4, 150.6 (C-4), 149.6 (C-3), 143.1 (Ar*CH*=CH), 128.1 (C-1), 122.1 (C-6), 113.4 (C-5), 112.6 (ArCH=*CH*), 110.5 (C-2), 77.4 (NOH*C*≡CH), 71.6 (NOHC≡*CH*), 68.3 (CH_2_O), 58.5 C*H*_2_CH_2_CH_2_CH_2_O), 55.1 (OCH_3_), 53.9 [2C, N(CH_2_)_2_], 37.9 (*CH_2_*C≡CH) 26.9 (C*H_2_*CH_2_O), 24.8 [2C, N(CH_2_)_2_(*CH_2_*)_2_)], 23.6 [N(CH_2_)_2_(CH_2_)_2_*CH_2_*], 22.5 (C*H*_2_CH_2_CH_2_O). HRMS (ESI) Calcd. for C_22_H_30_N_2_O_4_: 386.2204. Found 386.2206.

*N-Hydroxybenzamide (**15**).* A solution of benzoic acid (0.5 g, 4.10 mmol) in dry DMF (7 mL) was reacted with HATU (1.56 g, 4.10 mmol) and DIPEA (1.8 mL, 10.25 mmol). The reaction mixture was stirred at rt for 15 min. Next, hydroxylamine hydrochloride (0.28 g, 4.10 mmol) was added. After complete reaction, workup, purification via chromatography and recrystallization, compound **15** (0.252 g, 45%) was obtained as a solid: mp 120–123 °C; IR (KBr) *v* 3296, 3058, 2748, 1645, 1612, 1561, 1490, 1436, 1328, 1162 cm^−1^; ^1^H NMR (400 MHz, DMSO-*d_6_*) δ 11.21 (s, 1H), 9.03 (s, 1H), 7.77–7.73 (m, 2H), 7.52–7.43 (m, 3H). ^13^C NMR (101 MHz, DMSO-*d_6_*) δ 164.68 (s), 133.25 (s), 131.58 (s), 128.83 (s), 127.31 (s). HRMS (ESI) Calcd. for C_7_H_7_NO_2_: 137.0477. Found: 137.0478. 

*N-Hydroxy-N-(prop-2-yn-1-yl)benzamide (**16**).* A solution of benzoic acid (0.5 g, 4.10 mmol) in dry DMF (7 mL) was reacted with HATU (1.56 g, 4.10 mmol) and DIPEA (1.8 mL, 10.25 mmol). The reaction mixture was stirred at rt for 15 min. Next, *N*-(prop-2-yn-1-yl)hydroxylamine hydrochloride (0.44 g, 4.10 mmol) and an extra equivalent of DIPEA (0.71 mL, 4.10 mmol) were added. After complete reaction, workup, purification via chromatography, and recrystallization, compound **16** (0.332 g, 28%) was obtained as an amorphous solid: IR (KBr) *v* 3285, 2931, 2857, 2122, 1599, 1572, 1448, 1340, 1265, 1157 cm^−1^. ^1^H NMR (400 MHz, DMSO-*d_6_*) δ 10.28 (s, 1H), 7.65 (d, *J =* 7.0 Hz, 2H), 7.55–7.34 (m, 3H), 4.42 (d, *J =* 2.2 Hz, 2H), 3.25 (t, *J =* 2.2 Hz, 1H). ^13^C NMR (101 MHz, DMSO-*d_6_*) δ 169.72 (s), 134.52 (s), 130.97 (s), 128.83 (s), 128.29 (s), 79.50 (s), 74.84 (s). HRMS (ESI) Calcd. for C_10_H_9_NO_2_: 175.0633. Found: 175.0626.

### 3.2. Biological Evaluation: Inhibition of hChEs and hMAOs

Compounds **4**–**6**, **15**, and **16** were assayed for their inhibition of hChEs and hMAOs (all obtained from Sigma Aldrich). Inhibition assays were performed according to previously published protocols [24]. All compounds were screened at a concentration of 10 μM, and IC_50_ values were calculated for those showing inhibition *>*60%, by testing seven concentrations in the range 30–0.01 μM. Briefly, the classical Ellman spectrophotometric assay (for ChEs) [24] and fluorimetric detection of 4-hydroxyquinoline (for MAOs) [38] were adapted to a plate reading procedure using 96-well microtiter plates (Greiner Bio-Frickenhausen, Frickenhausen, Germany). Readings were performed using the Infinite M1000 Pro plate reader (Tecan, Cernusco s.N., Cernusco sul Naviglio, MI, Italy), and statistical regression was performed using Prism software (GraphPad Prism version 5.00 for Windows, GraphPad Software, San Diego, CA, USA). 

### 3.3. Molecular Modeling

The structure of compound **4** was prepared as a hydrochloride salt using standard bond lengths and angles with Discovery Studio, software package, version 2.1. The molecular geometry was energy-minimized with the CHARMm force field [39] using the Newton–Raphson algorithm, with a convergence criterion for the energy gradient of 0.01 kcal(mol-Å)^−1^ [40]. The ligand was set up for docking using AutoDockTools (ADT; version 1.5.4) to define the torsional degrees of freedom that should be considered during the docking process. All acyclic dihedral angles of the ligand were allowed to rotate freely. The three-dimensional coordinate of the enzyme was taken from the Protein Data Bank (PDB): hAChE in complex with fasciculin II (PDB entry 1B41).

To prepare the protein, all water molecules, heteroatoms, all co-crystallized solvents, and the co-crystalized ligand were removed. The correct bonds, bond orders, hybridizations, and charges were assigned using the protein model tool in the Discovery Studio software package, version 2.1. The CHARMm force field was applied using the receptor-ligand interaction tool of the Discovery Studio software package, version 2.1. The prepared protein was loaded directly into ADT, and the hydrogen atoms and partial charges for proteins and ligands were calculated using Gasteiger charges. To give flexibility to the binding site, eight residues lining the AChE active site (i.e., Trp286, Tyr124, Tyr337, Tyr341, Tyr72, Asp74, Thr75, Trp86) were allowed to move. AutoDock Vina software [25] was used for protein-ligand calculations, and the corresponding grid box was created with a resolution of 1 Å and 60 × 60 × 72 points and positioned at the center of the protein (x = 116.546; y = 110.33; z = −134.181). The docking box indicated by ADT formed a large region that encompassed the entire target protein (blind docking). The resulting structure files were analyzed using Discovery Studio.

### 3.4. DPPH Radical-Scavenging Assay

Compound **4** at concentrations of 3.125–50 µM was incubated for 1.5 h with 2,2-diphenyl-1-picrylhydrazyl radical (DPPH, 70 µM) in MeOH, and absorbance was measured (λ = 517 nm) using a Synergy™ H4 reader (BioTek Instruments, Inc., Winooski, VT, USA) [41]. The experiment was performed in three technical replicates, and the blank sample (compound **4** without DPPH) was subtracted. The results are expressed as the percentage of DPPH free radicals (DPPH free radicals (%) = ((*A*_0_ − *A*_1_)/*A*_0_) × 100, where *A*_0_ is the absorbance of DPPH in MeOH and *A*_1_ is the absorbance of DPPH in the presence of compound **4**). Resveratrol and Trolox were used for comparison.

### 3.5. Metal Chelation 

Metal chelation was determined in assay buffer (20 mM Na-HEPES, 150 mM NaCl, pH 7.4) according to the procedure described previously [42]. Stock solutions of the metals at 100 mM concentrations were prepared in MilliQ water (Milli-Q advantage A10 Ultrapure Water Purification System, Millipore, KGaA, Darmstadt, Germany). The FeCl_2_ stock solution (100 mM) was prepared in the presence of ascorbic acid (1 mM). Compound **4** at a concentration of 30 μM was incubated for 30 min with equimolar concentrations of CuCl_2_, ZnCl_2_, MgCl_2_, CaCl_2_, MnCl_2_, FeCl_2_, CoCl_2_, and NiCl_2_. Absorbance spectra were measured on a Synergy™ H4 reader (BioTek Instruments, Inc., Winooski, VT, USA).

## 4. Conclusions

AD is a neurodegenerative disorder and the most common form of senile dementia. The four current AD pharmacotherapies focus on affecting the cholinergic and glutamatergic systems, but with limited therapeutic success. Therefore, there is an urgent need to explore all avenues to combat this disease, such as the discovery of new scaffolds that bind to different proteins involved in AD to provide greater structural diversity and new solutions in this field. In this work, we described *N*-hydroxyamide hybrids **1**–**3** and *N*-hydroxy[*N*-(prop-2-yn-1-yl)]amide hybrids **4**–**6**, which were formed from the juxtaposition of Contilisant and FA, with the piperidine moiety attached to the oxygen in the C4 position of ferulic acid and the carboxylic acid converted into an amide with a propargyl motif. Among the synthesized compounds, we identified compound **4**, which showed micromolar inhibition of hAChE (IC_50_ = 2.63 μM) and antioxidant activity with an EC_50_ value of 40.9 ± 0.5 µM, a value significantly lower than that of resveratrol (EC_50_ = 50.1 ± 4.6 µM) but 3.3-fold higher than that of Trolox (EC_50_ =12.4 ± 0.3 µM). *N*-Propargylhydroxamate **4** chelated divalent copper ions, which are at the center of vicious oxidative stress, A*β* aggregation, and various other AD-related pathological changes, as well as iron Fe^2+^ ions, which is not surprising since Desferrioxamine B, a hydroxamate-based siderophore, also efficiently binds ferrous and ferric ions. With its favorable drug-like properties and predicted crossing of the BBB, compound **4** provides a solid basis for further optimization and development of anti-AD ligands. In summary, and with reference to SAR, we conclude that: (1) among hydroxamates **1**–**3**, where the length of the linker connecting the piperidine to the phenyl ring varies, the most potent derivative was analogue **3** with an *n*-butyl linker; (2) among *N*-propargylhydroxamates **4**–**6**, ligand **4** with a shorter, ethyl spacer was the most potent inhibitor of hAChE; and (3) *N*-propargylhydroxamate **4**, the most potent AChE inhibitor, did not inhibit MAOs.

In conclusion, the preliminary results reported here pave the way for the exploration and development of new *N*-propargylhydroxamates from natural products.

## Data Availability

The data presented in this study are available in Appendix A.

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
