# Peer review of "N-Hydroxy-N-Propargylamide Derivatives of Ferulic Acid: Inhibitors of Cholinesterases and Monoamine Oxidases"

_molecules, 2022, doi:10.3390/molecules27217437_

Round 1

Reviewer 1 Report

Comments in the attached pdf.

Author Response

1st Reviewer

 Review: “N-Hydroxy-N-propargylamide Derivatives of Ferulic Acid: Inhibitors of Cholinesterases and Monoamine Oxidases” by Bautista-Aguilera et al.

Authors, Bautista-Aguilera et al, present here a synthesis and evaluations of 6 compounds for an application as Alzheimer’s disease drugs. They first describe the complete synthesis of compounds in clear and relevant procedures. Then, they performed the inhibition evaluation as a short screening to select the active compounds. These results highlighted one hit: compound 4. This hit was then selected for the molecular docking study to understand and predict its binding modes to hAChE. Finally, authors explored the antioxidant and metal chelating properties of compound 4. In a general manner, the manuscript is well written. The purpose of the subject is clearly expressed and results are well redacted and argued. The authors’ work is clearly an innovative one for the research against the Alzheimer’s disease: Thank you very much for these kind comments. Although some things need to be improved or corrected, I highly recommend the publication of this article after some minor revisions.

Comments of the manuscript:

As Ferulic Acid (FA) is the staring material of the entire study, its sourcing is a crucial point. Indeed, pharmaceutical industry is still a pollutant one, using petroleum-based molecules. So, finding and using bio-sourced starting materials is clearly a turn to take, as well as for research groups as for Industry. Today, FA and derivatives can be bio-sourced and authors should mention this in their introduction (lines 82-86) https://doi.org/10.3389/fchem.2022.886367 https://doi.org/10.1002/cssc.202002141: Thanks for this suggestion, we incorporated the following text and new references 18 and 19:

“On the other hand, unlike petroleum-based molecules, natural products such as ferulic acid (FA) (Figure 1) are a biologically derived and sustainable material that is preferred in the chemical and pharmaceutical industries [18,19]. Moreover, they...”

[18] Rioux, B.; Combes, J.; Woolley, J. M.; Rodrigues, N. d. N.; Mention, M. M.; Stavros, V. G.; Allais, F From biomass-derived p-hydroxycinnamic acids to novel sustainable and non-toxic phenolics-based UV-filters: A multidisciplinary journey. Front. Chem. 2022, 10, 886367

[19] Flourat, A. L.; Combes, J.; Bailly-Maitre-Grand, C.; Magnien, K.; Haudrechy, A.; Renault, J.-H.; Allais, F. Accessing p-hydroxycinnamic acids: Chemical synthesis, biomass recovery, or engineered microbial Production? ChemSusChem 2021, 14, 118–129.

Line 100. There is a mistake in the scheme: authors identified the starting molecule as FA, but it is compound 7, please correct this: OK. Thanks and corrected.

Line 107. Scheme 4 caption: procedure a): the ester hydrolysis procedure should be the same as in Scheme 3, as it is written for compound 14 in lines 495-497. Please correct this and also check the yield as it is quite low compared to that of compounds 12 and 13: We thank the reviewer for noting this mistake, which was corrected accordingly. The yield for compound 14 is indeed lower compared to the yield of compounds 12 and 13, however we were not able to optimize it.

Line 205 and 220. Please add spaces for an easier read: Corrected.

Line 261-263. Could you moderate your argumentation about the EC50 of compound 4 vs Trolox? It is 3.3-fold higher than Trolox. However, you could highlight that EC50 of compound 4 is lower to that of Resveratrol: Yes. We have corrected this phrase as follows: “…N-Propargyhydroxamate 4 was an modest radical scavenger, showing an EC50 = 40.9 ± 0.5 µM, a value lower that of resveratrol (EC50 = 50.1 ± 4.6 µM), although 3.3-fold higher that of Trolox (EC50 =12.4 ± 0.3 µM)…”

Line 271-276. Question: why did you just limit your argumentation to the Cu2+ and Fe2+ ions? It is clear, even if it is less obvious, that the UV absorption is impacted in the presence of Ca2+ ions and it should be mentioned and discussed, even if it might be a concern about this chelation: The reviewer is right, a slight change of the spectra also occured in the presence of Ca2+ ions. Nontheless, the absence of isosbestic point gives a clear hint that this change of spectra was not a concequence of Ca2+/compound 4 complex formation.

The following sentence was added “A subtle spectral change in the presence of Ca2+ ions was also observed (Figure 4A), but the absence of isosbestic point indicates that this spectral change is apparently not due to the formation of Ca2+/4 complex formation.”

Reviewer 2 Report

Peer-Review Report (Reviewer Number 1)

October 21, 2022

Manuscript ID: 198046

Title: N-Hydroxy-N-propargylamide Derivatives of Ferulic Acid: In-2 hibitors of Cholinesterases and Monoamine Oxidases

Journal: Molecules

Overview of the manuscript

The manuscript describes the synthesis of six of ferulic acid derivatives, which are subjected to biological assessment targeting the inhibition of cholinesterase and monoamine oxidases. Those enzymes are the most potent target for neurodegenerative disorder disease as Alzheimer’s and Parkinson’s diseases. The author also studied molecular modeling, antioxidative, calculated ADME properties and metal chelating properties for the most potent compound.

Originality

The work accomplished is original and represent a continuous effort of previous published paper by the same group, especially the inspiration of the design of the target from Contilisant which already have a great potential to cross the BBB. The fusion of this high potential compound with FA a known molecule for its ability to promising pharmacological effects that could be useful for the treatment of Alzheimer’s and Parkinson’s diseases, such as antioxidant effects, neuroprotection, and its promotion of neurogenesis.

General recommendation:

The authors were inspired by the Contilisant, however this later was not used as reference to support the role played by the FA derivatives. On the top of that, other synthetized compound was not subjected to computational studies for a better understand of the SAR. The scoop for the synthetized compound lack off other candidates to discern the SAR like an aromatic ring bearing the piperidine as it the case with the N-propargyl hydroxamate.

Specific Comments

a)   Writing style and language: The manuscript is well written and the language throughout the manuscript is clear.

b)   Title: It is representing the work.

c)    keywords: Good chosen keywords

d)   Introduction:

The introduction section is faire and help the reader to understand the work by not being too much specific while being explaining the whole idea of the work.

e)      Scheme: It is better to reduce the size (schemes 2 and 4) by shrinking the linker to the piperidine using number (n=4) of carbons as for schemes 1 and 3.

f)      Results:

a.      Synthesis

 Line 134 to 137: The author stated “We were interested in this type … have the N-propargyl moiety that can irreversibly bind to the flavin-adenine dinucleotide cofactor of MAO enzyme”

However, they did not give information about the binding type or any reference to support this statement. A work earlier done by group explained how its bind [1].

b.     Biological Evaluation. Inhibition of hChEs/hMAOs.

The information concerning the source of the enzymes in line 147 should be included in the Materials and Methods section not in the discussion.

g)   Conclusion

The conclusion is involving a lot of general information which is already included in the introduction. Thus, other relevant result should be stated in this section as the metal chelation potential and the SAR conclusion.

References

1.        Samadi, A.; de los Ríos, C.; Bolea, I.; Chioua, M.; Iriepa, I.; Moraleda, I.; Bartolini, M.; Andrisano, V.; Gálvez, E.; Valderas, C., et al. Multipotent MAO and cholinesterase inhibitors for the treatment of Alzheimer's disease: Synthesis, pharmacological analysis and molecular modeling of heterocyclic substituted alkyl and cycloalkyl propargyl amine. European Journal of Medicinal Chemistry 2012, 52, 251-262, doi:https://doi.org/10.1016/j.ejmech.2012.03.022, URL: https://www.sciencedirect.com/science/article/pii/S0223523412001845.

Author Response

2nd Reviewer

Manuscript ID: 198046

Title: N-Hydroxy-N-propargylamide Derivatives of Ferulic Acid: Inhibitors of Cholinesterases and Monoamine Oxidases

Journal: Molecules

Overview of the manuscript

The manuscript describes the synthesis of six of ferulic acid derivatives, which are subjected to biological assessment targeting the inhibition of cholinesterase and monoamine oxidases. Those enzymes are the most potent target for neurodegenerative disorder disease as Alzheimer’s and Parkinson’s diseases. The author also studied molecular modeling, antioxidative, calculated ADME properties and metal chelating properties for the most potent compound.

Originality

The work accomplished is original and represent a continuous effort of previous published paper by the same group, especially the inspiration of the design of the target from Contilisant which already have a great potential to cross the BBB. The fusion of this high potential compound with FA a known molecule for its ability to promising pharmacological effects that could be useful for the treatment of Alzheimer’s and Parkinson’s diseases, such as antioxidant effects, neuroprotection, and its promotion of neurogenesis: Thank you very much for these kind comments.

General recommendation:

The authors were inspired by the Contilisant, however this later was not used as reference to support the role played by the FA derivatives. On the top of that, other synthetized compound was not subjected to computational studies for a better understand of the SAR. The scoop for the synthetized compound lack off other candidates to discern the SAR like an aromatic ring bearing the piperidine as it the case with the N-propargyl hydroxamate:

Reviewer is correct, however our aim was to incorporate only the pharmacophoric elements of Contilisant. Computational studies could indeed be expanded to other compounds, yet only the most potent inhibitor was taken into the consideration, since it is also the most interesting compound given its multifunctional profile of activities. Also, computational studies are commonly performed only on the most potent inhibitor. This paper can be seen as a starting point for further studies, and additional derivatives (as those suggested by the reviewer) will be synthesized and analysed in our laboratories.

Specific Comments

  1. a)Writing style and language: The manuscript is well written and the language throughout the manuscript is clear: Thank you very much for these kind comments.
  2. b)Title: It is representing the work: Thank you very much for this comment.
  3. c)keywords:Good chosen keywords: Thank you very much for this kind comment.
  4. d)Introduction:

The introduction section is fair and help the reader to understand the work by not being too much specific while being explaining the whole idea of the work: Thank you very much for these kind comments.

  1. e)Scheme: It is better to reduce the size (schemes 2 and 4) by shrinking the linker to the piperidine using number (n=4) of carbons as for schemes 1 and 3: Schemes 2 and 4 were modified as suggested by the reviewer.

  1. f)Results:
  2. Synthesis

 Line 134 to 137: The author stated “We were interested in this type … have the N-propargyl moiety that can irreversibly bind to the flavin-adenine dinucleotide cofactor of MAO enzyme”

However, they did not give information about the binding type or any reference to support this statement. A work earlier done by group explained how its bind [1]: We thank the reviewer for drawing our attention to this missing information. To support our claims ref. 16 (our previous work), where N-propargylamines are described and experimentally confirmed as irreversible inhibitors.

  1. Biological Evaluation. Inhibition of hChEs/hMAOs.

The information concerning the source of the enzymes in line 147 should be included in the Materials and Methods section not in the discussion: Corrected as suggested by the reviewer: the missing information was added to the Materials and Methods section.

  1. g)Conclusion

The conclusion is involving a lot of general information which is already included in the introduction. Thus, other relevant result should be stated in this section as the metal chelation potential and the SAR conclusion: As suggested by the reviewer, the Conclusion section was modified to include also other relevant information.

References

  1. Samadi, A.; de los Ríos, C.; Bolea, I.; Chioua, M.; Iriepa, I.; Moraleda, I.; Bartolini, M.; Andrisano, V.; Gálvez, E.; Valderas, C., et al. Multipotent MAO and cholinesterase inhibitors for the treatment of Alzheimer's disease: Synthesis, pharmacological analysis and molecular modeling of heterocyclic substituted alkyl and cycloalkyl propargyl amine. European Journal of Medicinal Chemistry 201252, 251-262, doi:https://doi.org/10.1016/j.ejmech.2012.03.022, URL: https://www.sciencedirect.com/science/article/pii/S0223523412001845

Round 2

Reviewer 2 Report

Thank you for addressing the comments and recommendations and also for your effort to improve the manuscript.